# New Prognostic Factors in Operated Extracapsular Hip Fractures: Infection and GammaTScore

**DOI:** 10.3390/ijerph191811680

**Published:** 2022-09-16

**Authors:** Carlos Hernández-Pascual, José Ángel Santos-Sánchez, Jorge Hernández-Rodríguez, Carlos Fernando Silva-Viamonte, Carmen Pablos-Hernández, Manuel Villanueva-Martínez, José Antonio Mirón-Canelo

**Affiliations:** 1Department of Human Anatomy and Histology, Faculty of Medicine, Campus Miguel de Unamuno, Universidad de Salamanca, Avda, Alfonso X el Sabio s/n, 37007 Salamanca, Spain; 2Department of Traumatology and Orthopaedic Surgery, Hospital Universitario de Salamanca, Pso. San Vicente 58-182, 37004 Salamanca, Spain; 3Department of Biomedical and Diagnostic Sciences (Area of Radiology and Physical Medicine), Faculty of Medicine, Campus Miguel de Unamuno, Universidad de Salamanca, Avda, Alfonso X el Sabio s/n, 37007 Salamanca, Spain; 4Department of Statistics, Faculty of Medicine, Campus Miguel de Unamuno, Universidad de Salamanca, Avda, Alfonso X el Sabio s/n, 37007 Salamanca, Spain; 5Department of Geriatrics, Hospital Universitario de Salamanca, Pso. San Vicente 58-182, 37004 Salamanca, Spain; 6Avanfi Institute, C/Orense 32, 28020 Madrid, Spain; 7Department of Preventive Medicine and Public Health, Faculty of Medicine, Campus Miguel de Unamuno, Universidad de Salamanca, Avda, Alfonso X el Sabio s/n, 37007 Salamanca, Spain

**Keywords:** hip fracture, surgery, fracture fixation, intramedullary, treatment outcome, complications, risk factor

## Abstract

There is no universal postoperative classification of extracapsular hip fractures (ECFs). We wondered if infection (according to infection after fracture fixation criteria (IAFF)), immediate partial weight bearing (PWB) and/or the new GammaTScore tool could predict early cut-out. We also examined the correlation between GammaTScore and time to consolidation and studied long-term survival. This was a retrospective cohort study of low-energy complete ECFs operated with Gamma3T nailing in 2014 and fully monitoring, in patients aged over 65. Ten not distally locked cases, one late cut-out, one cut-through, one osteonecrosis and one pseudarthrosis were discarded. Patients were classified into early cut-out (7/204; 3.55%) and no early cut-out (197/204; 96.45%). There was a lower percentage of A2 fractures according to the AO Foundation/Orthopaedic Trauma Association classification (AO/OTA, 1997) in early cut-out. IAFF and only the GammaTScore reduction parameter were different for early cut-out, in opposition to immediate PWB, tip-to-apex distance (TAD) or the Baumgaertner–Fogagnolo classification. GammaTScore inversely correlated with consolidation (*p* < 0.01). Long-term survival time was not statistically significantly lower in the early cut-out group. Small sample of cases may limit our results. Apart from an important role of IAFF, GammaTScore would be useful for predicting consolidation, avoiding complications and reducing costs. Further studies are needed for reliability.

## 1. Introduction

Extracapsular hip fractures (ECFs) in elderly patients have a high morbidity and mortality rate, which represents a serious international public health problem [1]. Most of these fractures are treated surgically. Confirming healing and minimizing complications is essential [2], as the clinical consequences of failure can be devastating [3]. ECFs treated with nailing almost always consolidate [4]. The cause for the approximately 5% that do not is mainly summarized as non-traumatic mechanical complications (NTMCs) [5].

The most common NTMC reported is cut-out. Parker first defined it in 1992 as the “projection of the lag screw from the femoral head by more than 1 mm”, referring to the dynamic hip screw (DHS) [6], but later on, it changed to “the collapse of the neck-shaft angle into varus, leading to extrusion of the screw from the femoral head“ (Baumgaertner, 1995) [7]. The compilation of cut-out cases presents three main difficulties:-Physiopathology: It is probably underdiagnosed when we only observe disrotation without migration. However, there are some cases when the cut-out does not progress and the fracture consolidates, but we do not know when and why this occurs [8];-Differential diagnosis: Cut-in [9] and cut-through [10] are still mistaken for cut-out;-Time of diagnosis: Bojan considered as early NTMCs those detected within 3 months postoperatively [11]. A recent article [12] includes them as “Early Mechanical Complications” (EMCs) if they occur before the first postoperative year. 

It seems that the cut-out rate experienced in fractures treated with short nails is the same as in those treated with long ones [13]. In addition, recent clinical works reject the theoretical superiority of blade versus screw as proximal locking devices (PLDs) [14]. Its incidence has progressively decreased from 16% to less than 8%, and it is currently 1.6–4.3% [15]. This phenomenon can probably be explained by the following reasons:-Design improvements [16]: Smaller nail diameters and advances in PLDs;-Learning curve [17]: The quick widespread use of nails in all ECFs (instead of DHSs);-Biomechanical factors: Cleveland and Bosworth [18] wrote that the PLD distal end in the centre–centre or posteroinferior quadrant prevents from this complication. Parker [6] pointed out that the ideal position was central or inferior in anteroposterior view and central in lateral view. There is no consensus on the peripheral position with worst prognosis [4]. Baumgaertner demonstrated that a < 25 mm tip–apex distance (TAD), as the sum of anteroposterior and axial projections, is a protective factor [7]. Subsequently, new factors or modifications of previous ones have been described, but they have not been as universally accepted or recognized [19]: calcar-referenced TAD < 25 mm [20], varus reduction [21], ECF extension to the femoral neck [22], posterior subtype of Ikuta’s classification [23], vertical shear fracture [24] and intraoperative breakage/lack of lateral wall competency [25].

Heretofore, postoperative infection has not been clearly implicated in early cut-out, but it is especially difficult to confirm really depth ones [26]. Moreover, there is a clear tendency to immediately allow patients to perform partial weight bearing (PWB) postoperatively, but there are many protocols, and not all patients collaborate, so the role this factor plays is undefined [27].

Preoperative classifications have only found that the DHS is not indicated in unstable and reverse fracture patterns [21,28]. The AO Foundation/Orthopaedic Trauma Association classification (AO/OTA 1997, updated in 2007), widely used in research [29], has poor inter- and intra-observer reliability. Massoud [30] has introduced the “basicervical-equivalent” concept, but it has not succeeded. Postoperatively, ECFs fractures are stable or unstable; classic unstable patterns include insufficient posteromedial cortex contact/comminution, subtrochanteric extension and/or reverse obliquity. Baumgaertner–Fogagnolo [31] classified them based solely on reduction. In 2013, another tool was developed, but with a complex scoring system [32].

We believe it is important to know the risk factors with prognostic value. We researched if IAFF (infection after fixation fracture), immediate PWB and/or the use of a new postoperative tool, GammaTScore, could predict early cut-out and if there was a correlation between early cut-out and time to consolidation. Finally, we focused on long-term survival in both complicated and non-complicated cases.

## 2. Materials and Methods

This study was performed at Hospital Universitario de Salamanca (HUS), which is a third-level university hospital and a regional centre of reference for certain services.

The design used to achieve the objective and test the hypothesis of this work consisted of a historical cohort study of ECF-diagnosed patients in our department who were over 65 years old in 2014. Our previous publication [33] outlines the sample collection, with inclusion and exclusion criteria, as well as the material and methods employed for consolidation definition, radiological views and measurements, and follow-up protocol and survival. All our patients were considered osteoporotic, given the recruitment.

We evaluated 204 titanium Gamma 3 trochanteric nails (hereafter Gamma3T)—length of 180 mm and distal width of 11 mm, with neck–shaft angle between 120° and 130° (Stryker^®^ Trauma GmbH, Schörnkirchen, Germany)—with a total of 7 cases called “Early cut-out” (3.55%) and 197 called “No early cut-out” (96.55%). Four major mechanical complications were excluded due to possible interference: one late cut-out, one cut-through, one pseudarthrosis and one osteonecrosis (Figure 1).

Early cut-out was defined as any grade of disrotation and/or migration of the PLD in the cephalic fragment, without (incomplete, grades I and II) or with (complete, grade III) articular damage, at any point throughout follow-up until 6 months postoperatively. All infections were considered as depth infections and met at least one of the confirmatory criteria for IAFF. Immediate postoperative PWB was authorized if reduction was considered good as per the Baumgaertner–Fogagnolo classification [31] criteria.

To elaborate GammaTScore, we empirically identified and classified most of the unfavourable prognostic variables into 3 parameters based on reduction, osteosynthesis and instability, with a score ranging from 1 to 3 points. The final rate of the 3 × 3 algorithm was considered good for 8–9 points, moderate for 5–7 points and poor for 3–4 points (Figure 2 and Figure 3).

Reduction parameter (R):
CCD (Varus/Neutral/Valgus): Difference between the caput–collum–diaphyseal (CCD) angle of the currently operated hip and that of the contralateral one (if the latter was previously operated on, this CCD was taken as a reference);AP%: Percentage of cortical contact in AP view;P, N or A: Posterior, normal or anterior types (Ikuta’s classification, axial view).Osteosynthesis parameter (O):TAD: Tip-to-apex distance;PLD: Location of the proximal locking device (PLD) in the femoral head (Cleveland–Bostworth classification).Instability parameter (I):Avulsion: Defined as a radiolucent space > 5 mm in any radiological view of the greater trochanter (GT) or the lesser trochanter (LT);LW (lateral wall) in/competence (AO Foundation/Orthopaedic Trauma Association classification; AO/OTA classification 2018).

Descriptive statistics generated using SPSS 20.0 (SPSS, Inc., Chicago, IL, USA) were utilized for data analysis. Kolmogorov–Smirnov’s normality tests were used to check if the continuous variables matched a normal distribution, and comparisons were performed with Mann–Whitney’s U tests. For categorical variables, Pearson’s chi-square tests and Fisher’s exact tests were used, and the median test was used to calculate discrete continuous variable risk score. All *p*-values were two-sided, and *p*-values below 0.05 were considered significant. Mantel–Cox’s log-rank, Breaslow’s (generalized Wilcoxon) and Tarone-Ware’s tests were used to evaluate survival, yielding no significant differences. Boxplots and bar diagrams were generated to compare the main outcomes.

## 3. Results

### 3.1. Preoperative Variables (Table 1)

Cohorts were similar, including social situation, dependency (Barthel’s index [34]), comorbidity (Charlson’s comorbidity index, CMI [35]), cognitive impairment (Pffeifer’s classification [36]), osteoporosis (OP; Nuti’s classification [37]), previous OP treatment, anti-platelet therapy/anti-coagulation therapy (APT/ACT) and ASA score [38].

**Table 1 ijerph-19-11680-t001:** Preoperative variables. Tests: a, Fisher’s exact test; b, Pearson’s chi-square test; c, Mann–Whitney’s U test; d, median test (Sx., surgery; CMI, Charlson’s comorbidity index; OP, osteoporosis; Fx., fracture; APT, anti-platelet therapy; ACT, anti-coagulation therapy; AAS, acetylsalicylic acid; LMWH, low-molecular-weight heparin; ASA, American Society of Anesthesiologists). ^#^ Upon admission, no calcium and/or vitamin D.

Variable	Early Cut-Out	No Early Cut-Out	*p*
Age (years-old)	83 (SD 10.31) (65;99)	85.82 (SD 6.59) (65;103)	0.512 c
Sex			0.640 a
-Female	5 (71.4%)	156 (79.2%)	
-Male	2 (28.6%)	41 (20.8%)	
Side			1.000 a
-Left	3 (42.4%)	81 (41.1%)	
-Right	4 (57.6%)	116 (58.9%)	
Social situation			0.121 a
-Home, alone	3 (42.9%)	26 (13.2%)	
-Home, not alone	3 (42.9%)	104 (52.8%)	
-Institutionalized	1 (14.3%)	67 (34.0%)	
Dependency			
-Barthel pre-Sx.	86.43 (SD 16.51) (55;100)	75.28 (SD 21.57) (10;100)	0.150 c
-Barthel post-Sx.	43.57 (SD 21.74) (15;75)	46.78 (SD 20.93) (10;85)	0.697 c
Comorbidity			
-Non-age-adjusted CMI	2.57 (SD 1.13) (1;4)	2.15 (SD 1.29) (0;7)	0.293 c
-Age-adjusted CMI	6.42 (SD 1.27) (5;8)	6.11 (SD 1.30) (4;11)	0.467 c
Cognitive status			1.000 a
-None	1 (14.3%)	33 (16.8%)	
-Mild	5 (71.4%)	130 (66.0%)	
-Moderate	1 (14.3%)	33 (16.8%)	
-Severe	0 (0%)	1 (0.5%)	
Previous Fx.			1.000 a
-None	5 (71.4%)	114 (57.9%)	
-Traumatic	0 (0%)	6 (3.0%)	
-OP fracture			
--Hip	0 (0%)	15 (7.6%)	
--Other locations	2 (28.6%)	54 (27.4%)	
--Both locations	0 (0%)	8 (4.1%)	
OP treatment ^#^			0.596 b
-No	129 (85.4%)	47 (82.5%)	
-Yes	22 (14.6%)	10 (17.5%)	
APT/ACT			0.882 a
-None	5 (71.4%)	110 (55.8%)	
-AAS 100	2 (28.6%)	39 (19.8%)	
-AAS 300	0 (0%)	17 (8.6%)	
-Clopidogrel	0 (0%)	1 (0.5%)	
-Acenocumarol	0 (0%)	26 (13.2%)	
-Direct Xa	0 (0%)	2 (2.0%)	
-LMWH	0 (0%)	2 (2.0%)	
ASA	3 (1;4)	3 (1;4)	0.503 d

### 3.2. Perioperative Variables (Table 2)

There were only significant differences in the AO/OTA classification (1997) [29] (less A2 cases), but not in Jensen [28] or Massoud [30], nor in the medical variables.

**Table 2 ijerph-19-11680-t002:** Perioperative variables. Tests: a, Fisher’s exact test; b, Pearson’s chi-square test; c, Mann–Whitney’s U test (AO/OTA, AO Foundation/Orthopaedic Trauma Association classification; Sx., surgery; [RBC], red blood cells concentrate). *, Statistically significative.

Variable	Early Cut-Out	No Early Cut-Out	*p*
Jensen			0.345 a
Stability (Jensen)			0.713 a
-Stable (I + II)	3 (42.9%)	73 (37.1%)	
-Unstable (III + IV + V)	4 (57.1%)	124 (62.9%)	
AO/OTA, 2007 *			0.033 b, *
-A1	3 (42.9%)	59 (29.9%)	
-A2	1 (14.3%)	111 (56.3%)	
-A3	2 (28.6%)	17 (8.6%)	
Stability(AO/OTA, 2007)			0.305 b
Stable	3 (42.9%)	118 (59.9%)	
Unstable	3 (42.9%)	69 (35.0%)	
Basicervical (B2.1)	1 (14.3%)	10 (5.1%)	
Massoud			1.000 b
-Stable	4 (57.1%)	68 (34.5%)	
-Unstable	3 (42.9%)	129 (65.5%)	
Average stay (days)	9.57 (SD 2.63) (6;13)	9.71 (SD 3.36) (4;26)	0.880 c
Pre-Sx. stay (days)	3.00 (SD 2.00) (0;5)	3.30 (SD 2.56) (0;9)	0.749 c
Post-Sx. stay (days)	6.57 (SD 2.22) (3;9)	6.37 (SD 2.79) (3;25)	0.431 c
Blood loss (g/dl Hb)	1.42 (SD 1.16) (0;3.0)	1.92 (SD 1.75) (−3.5;5.9)	0.323 c
Transfusions [RBC]	1.85 (SD 1.67) (0;5)	1.35 (SD 1.37) (0;8)	0.427 c

### 3.3. Postperative Variables

IAFF was associated with a higher risk of early cut-out. We did not detect differences in TAD, Baumgaertner–Fogagnolo classification [31] or immediate PWB (Table 3).

Only the reduction parameter was statistically significant, whereas the osteosynthesis and instability parameters and the GammaTScore final rate did not show any differences (Figure 4).

### 3.4. Consolidation (Figure 5)

We observed a highly significant correlation with GammaTScore, with an R2 value of 0.108.

**Figure 5 ijerph-19-11680-f005:**
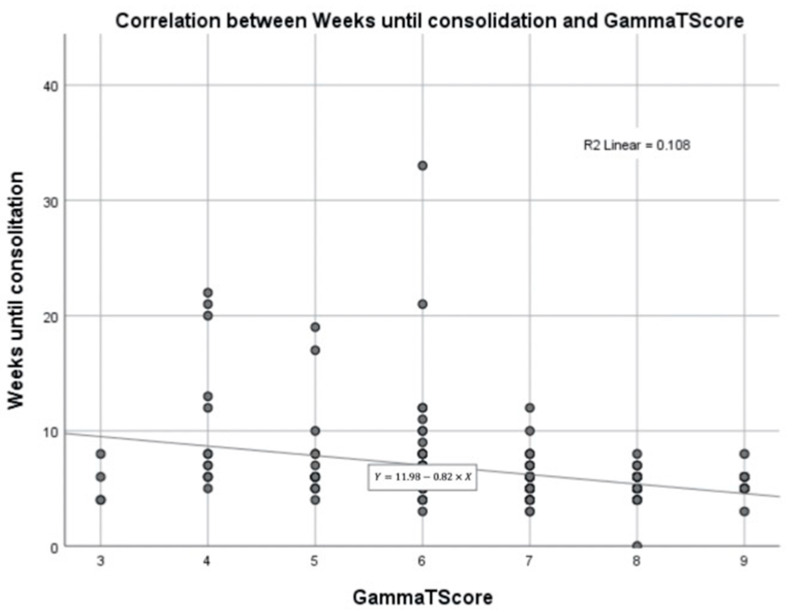
Correlation between weeks until consolidation and GammaTScore.

### 3.5. Survival (Figure 6)

Long-term survival was similar in both groups (*p* =  0.518) after almost 6 years of follow-up.

**Figure 6 ijerph-19-11680-f006:**
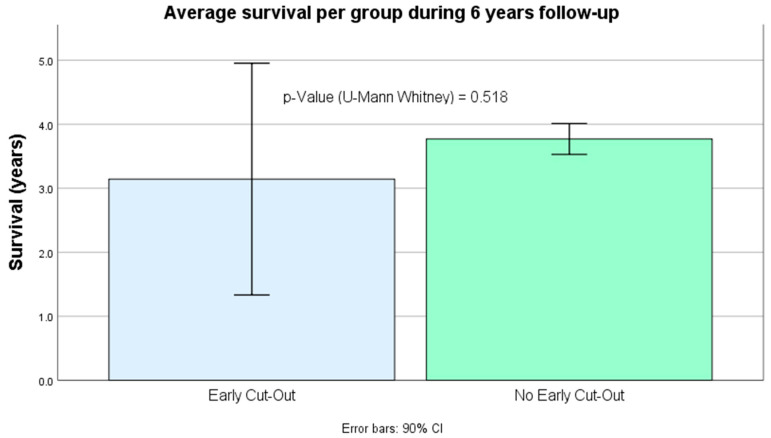
Average survival per group during almost 6 years follow-up.

### 3.6. Summary of Evolution of Early Cut-Out Cohort (Table 4)

Of the total of seven early cut-out cases, four cases were considered complete, and three cases were considered incomplete. For their resolution, prosthesis implantation was chosen in three cases, the removal of the material in three others (one consolidated) and observation in the remaining one, which consolidated without progressing.

**Table 4 ijerph-19-11680-t004:** Summary of the early cut-out cohort. (M, male; F, female; AO/OTA, AO Foundation/Orthopaedic Trauma Association classification; PWB, partial weight bearing; IAFF, infection after fracture fixation; ROM, removal of osteosynthesis material; PHA, partial hip arthroplasty; THA, total hip arthroplasty).

No.	Sex/(Age)	Jensen Stability		AO/OTA (1997) Stability			Massoud Stability		Immediate PWB	Detection Grade (Weeks)	IAFF (Weeks)	Consolidation/Reintervention (Weeks)
		Yes	No	Yes	No	2.1	Yes	No				
**1**	M (88)	**√**	**×**	**√**	**×**	**×**	**×**	**√**	Yes	III (4.7)	Yes (10.0)	No/ROM (10.7)
**2**	F (88)	**×**	**√**	**×**	**√**	**×**	**×**	**√**	No	III (23)	Yes (0.71)	No/ROM (25.7)
**3**	F (79)	**×**	**√**	**×**	**√**	**×**	**√**	**×**	No	II (13)	Yes (29.4)	Yes/ROM (30.0)
**4**	M (77)	**×**	**√**	**×**	**√**	**×**	**√**	**×**	No	II (3.0)	No	No/PHA (3.7)
**5**	F (65)	**√**	**×**	**×**	**×**	**√**	**√**	**×**	No	III (5.0)	No	No/THA (16.4)
**6**	F (87)	**×**	**√**	**√**	**×**	**×**	**×**	**√**	No	I (4.3)	No	Yes/-
**7**	F (97)	**√**	**×**	**√**	**×**	**×**	**√**	**×**	Yes	III (2.1)	No	No/PHA (2.86)

## 4. Discussion

ECF surgery is a very frequent and standardized procedure that occupies a large part of our activity. Although it has traditionally been considered to be relatively easy, ideal for residents in their early stages to perform, its complications are extremely serious, given that the vast majority occur in elderly patients with associated comorbidity. As a consequence, in recent years, some publications have been released that warn of the often fatal clinical consequences of those cases that require reintervention [39], as well as others that standardize every surgical step [40] in order to avoid errors in osteosynthesis.

For many years, OP has been postulated as a theoretical risk factor or even a confounding factor. In patients with hip fracture, its study using densitometric criteria is practically unfeasible [41]. Consequently, the current World Health Organization (WHO) classification [42], which is based on the severity of osteopenia severity in densitometry, has been criticized for years. Accordingly, we opted for Nuti’s classification [37], as it is simpler and easier for discriminating OP grades, which were similar in both cohorts. Elliot-Gibson stated that barely a third of osteoporotic patients are prescribed antiresorptive/osteoformative therapy and that, of these, less than a third complete the entire treatment [43]. In our study, less than 20% were treated in both groups, with no differences.

We did not find any previous work indicating confirmed IAFF [26] as a risk factor for early cut-out, which we explain it as being caused by the difficult follow-up, its apparently easy control or resolution with antibiotherapy or its insufficient registration (they are usually diagnosed and treated by primary care physicians). Note that in two complicated cases, IAFF occurred weeks later, which makes suspicion even more relevant. Bojan [8] only observed it in some cases of late cut-out. We believe that it is necessary to draw attention to any infectious complication of the surgical wound, which should be always registered in the medical records, using standardized criteria.

There is a clear trend among multiple protocols towards endorsing immediate PWB [44], regardless of whether patients are really able to comply with it [5]. However, its benefit/risk ratio has not actually been proven for intracapsular fractures, and we did not find any article on effective load and early cut-out. In our case, this variable was approved in a similar proportion in both groups. This argues for its limited role in the main NTMC, although the low power of our study opens the door to ulterior investigation.

Current classifications of ECFs are still based on preoperative radiological images, and do not include postoperative aspects that may influence the final outcome, such as obtained reduction [45], PLD osteosynthesis [46] or instability factors [25]. All these reasons motivated the creation of the GammaTScore as an immediate and simple postoperative prognostic tool that could help us reducing the costs involved.

Its reduction parameter showed differences, confirming that it appears to be more crucial than the rest [5]. Ikuta’s classification [23] enhanced it, and fracture varization has recently been postulated as fundamental [47]. Although it was already first intuited by Baumgaertner and Fogagnolo [31], the new tool seems to be more sensitive.

The GammaTScore osteosynthesis parameter was statistically similar, despite TAD being greater than 25 mm in the early cut-out group. Limited TAD and centre–centre positioning have been incorporated since the 1990s and are generally accepted in routine practice, as they are in our centre. Their minor role is probably due to minimal absolute differences in PLD procedures between cut-out and non-cut-out cases, as occurs in our work, which could explain the fact that these variables are currently being challenged [17].

Similarly, the instability parameter result was not statistically significant. Since the avulsion of the lesser trochanter has already been described as a classic instability factor [48], this work incorporated it postoperatively. The same occurred for the greater trochanter, which is clearly related to lateral wall incompetence, admitted in the latest AO/OTA classification [49]. Probably the small sample size of cases and few existing differences explain this outcome.

Both the Baumgaertner–Fogagnolo classification and the final score of GammaTScore were not sensitive enough, even though the former is focused on reduction. Assuming that increased attention to the surgical outcome led to more uniform radiological results, it was really difficult to establish any relevant differences. Anyway, the proportion of early cut-out seemed to be stabilized in less than 5% of cases, which was in line with our cumulative incidence (3.55%); therefore, other non-mechanical risk factors could have been behind them.

We also demonstrated an inverse relationship between GammaTScore and time to consolidation, which could serve for better patient management in office. However, data must be interpreted with caution because they only explained nearly 11% of the total.

Although long-term survival in early cut-out cases was lower, we paradoxically did not find a significant relationship. Less aggressive secondary surgeries, and even conservative final treatment in one case, are possibly sufficient to justify this finding.

We used a highly selected sample and neutralized many of the confounding factors involved, with a long, systematic follow-up that allowed us to evaluate survival. Our study also had some weaknesses, especially the small number of early cut-out cases and its retrospective, non-randomized design that might limit results and reliability.

## 5. Conclusions

Besides the new role description of IAFF in the main NTMCs of ECFs, the GammaTScore prognostic tool could be useful for preventing early cut-out, which often implies ulterior and severe reinterventions. Moreover, it could provide an easier monitoring prediction until consolidation. All of this might reduce important associated costs. In addition, if proven reliable, it could be applied to other single-PLD-based osteosyntheses.

## Figures and Tables

**Figure 1 ijerph-19-11680-f001:**
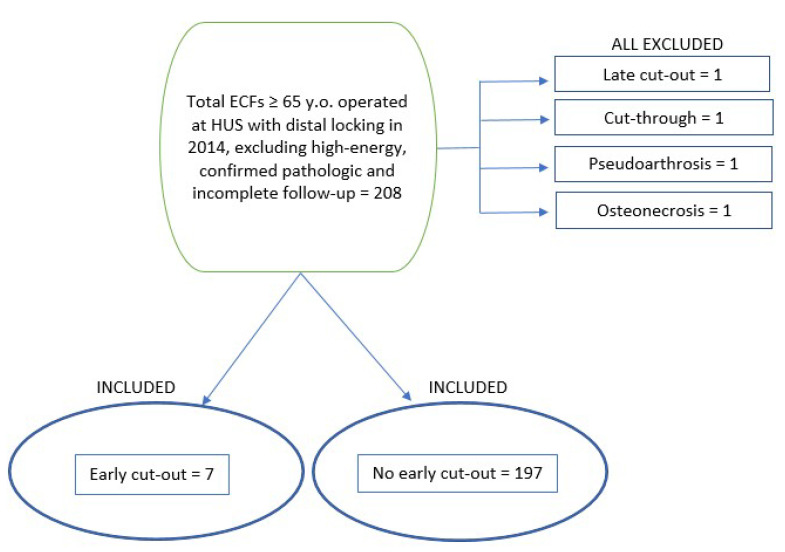
Organization chart (ECF, extracapsular fracture; HUS, Hospital Universitario de Salamanca; Sx., surgery.).

**Figure 2 ijerph-19-11680-f002:**
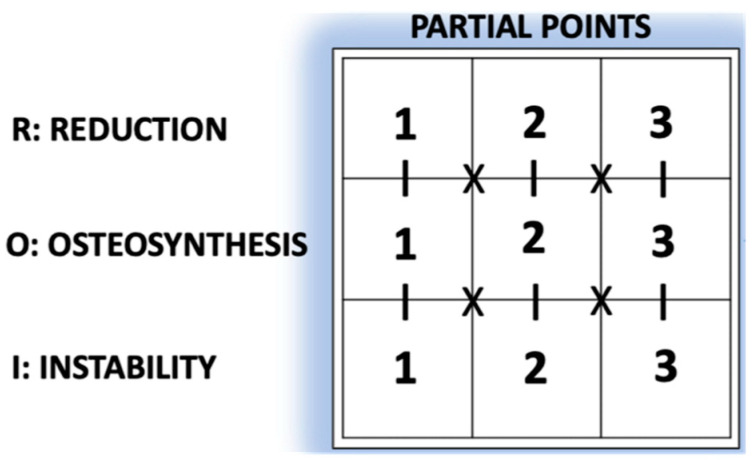
GammaTScore scoring system.

**Figure 3 ijerph-19-11680-f003:**
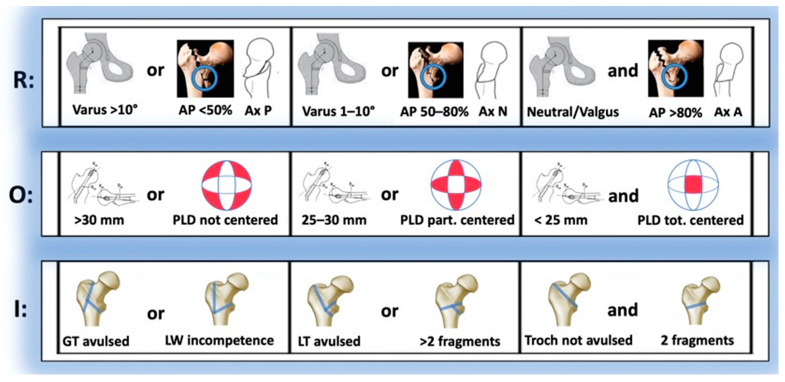
GammaTScore algorithm obtention chart.

**Figure 4 ijerph-19-11680-f004:**
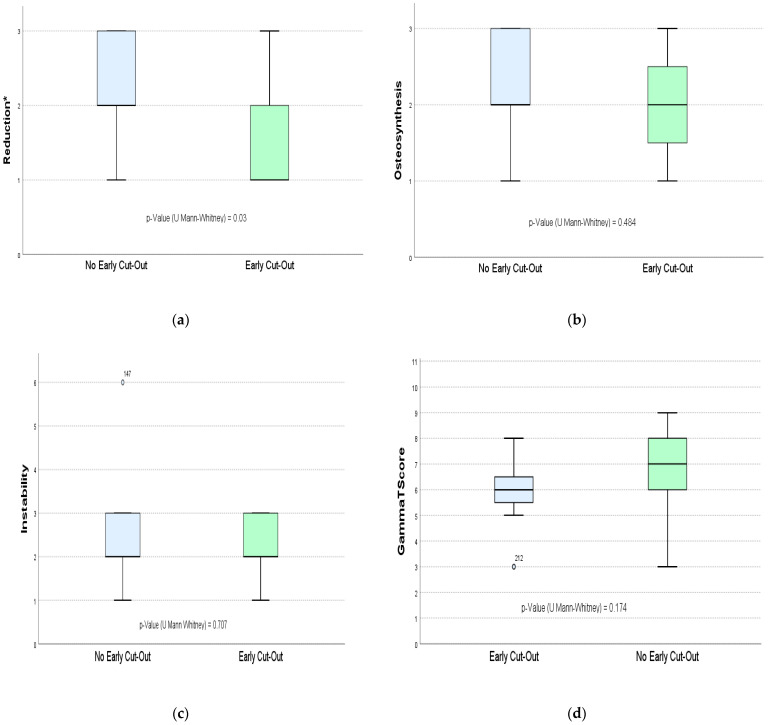
Representation of results of the GammaTScore tool, from top to bottom and from left to the right: (**a**) GammaTScore parameter—reduction; (**b**) GammaTScore parameter—osteosynthesis; (**c**) GammaTScore parameter—instability; (**d**) GammaTScore final rate. *, Statistically significative.

**Table 3 ijerph-19-11680-t003:** Tests: a, Fisher’s exact test; c, Mann–Whitney U test (TAD, tip-to-apex distance; IAFF, infection after fracture fixation; Sx., surgery; PWB, partial weight bearing). *, Statistically significative.

Variable	Early Cut-Out	No early Cut-Out	*p*
Number of patients	7	197	
TAD	25.54 (SD 8.81)	22.67 (SD 6.44)	0.395 c
Baumgaertner–Fogagnolo			
Poor	1 (14.3%)	7 (3.6%)	0.072 a
Moderate	4 (57.1%)	64 (32.5%)	
Good	2 (28.6%)	126 (64.0%)	
Infection (IAFF) *			0.002 a, *
-No	4 (57.2%)	191 (96.9%)	
-Yes	3 (42.8%)	6 (3.1%)	
Immediate post-Sx. PWB			0.451 a
-No	5 (71.4%)	103 (52.8%)	
-Yes	2 (28.6%)	94 (47.2%)	

## Data Availability

The data presented in this study are available upon request from the corresponding author.

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
