# Peer review of "New Prognostic Factors in Operated Extracapsular Hip Fractures: Infection and GammaTScore"

_ijerph, 2022, doi:10.3390/ijerph191811680_

Round 1

Reviewer 1 Report

Please see Editorial comments

Author Response

Comments to Review Report 1:

English language and style: English language and style are fine/minor spell check required

In this regard, we have shortened and revised the whole text again with the help of a professional translator (for the second time) and improved its style, making it more attractive and easier to read, with the addition of 2 more figures and the removal of one table.

Questions:

Does the introduction provide sufficient background and include all relevant references? Must be improved.

The introduction has been notably shortened and the bibliographic citations have been revised (see below).

Are all the cited references relevant to the research? Must be improved

We have checked-up all the references again. We have made various changes to make introduction more consistent, adding also one reference of IJERPH related to PWB. We have eliminated some references in Results, which we believe does not affect the integrity of the article itself. To sum up, the total number of references remains almost the same (49 now versus 50 before), except new recommendation by your revision.

Is the research design appropriate? Must be improved

We agree with the reviewer that the research design could have been better, especially if we had had more early cut-out cases, but recruitment is tremendously difficult and hard in such an elderly population. This circumstance has forced us to use statistical tests for non-normal samples such as the Mann-Withney U test in the study of our prognostic tool. However, the incidence of early cut-out in our study is within the range of that reported in the literature, as we state in Discussion.

For more clarity, we include here the results of the normality tests for the variables shown in the subsequent bar charts, as a justification for the use of Mann-Whitney U test(p-value of K-S tests for normality).

Variable p (No Early Cut-out) / p (Early Cut-out)

-TAD 0.003 / 0.200

-Reduction 0.000 / 0.015

-Osteosynthesis 0.000 / 0.200

-Instability 0.000 / 0.182

As p<0.05 for the No Early cut-out group, the null hypothesis is rejected: there is no evidence that the data follow a normal distribution.

As p>0.05 for the Early cut-out group, the null hypothesis is not rejected: they follow a normal distribution (although the significance values are not very high).

However, the incidence of early cut-out in our study is within that reported in the literature, as we state in Discussion (less than 4%).

Our sample has important follow-up and homogeneity, controlling confusion factors like osteoporosis, and our results are consistent with the new literature findings that support the role of IAFF and an increasingly important fracture reduction versus TAD, possibly because the latter is so implanted in surgeons that it is difficult to find relevant differences between cases and controls. 

Are the methods adequately described? Must be improved

Regarding this issue, the authors have preferred to shorten the text and refer the reader to reference number 33, which explains in detail how clinical and radiological variables were obtained, and how follow-up was implemented. Nevertheless, key points such as early cut-out have been pointed out or presented as well as confirmatory criteria for IAFF diagnosis.

Are the results clearly presented? Must be improved

The authors have made a considerable effort to organize the results and present them in an attractive way, minimizing the use of long tables and long paragraphs. The inclusion of discrete variables for GammaTScore marking has limited the use of other graphics, but our commitment to this punctuation system facilitates an easy use of the tool for an immediate result, which it is really interesting in our busy clinical offices.

Are the conclusions supported by the results? Must be improved 

Conclusions have been reformulated and shortened for better understanding and clarity for the reader.

Comments and Suggestions for Authors

Please see Editorial comments.

The authors have read Editorial comments and have replied to them below.

Please use the version of your manuscript found at the above link for your revisions. 

The authors have used the referred version to make the corrections.

(I) Please check that all references are relevant to the contents of the manuscript.

We have checked up all the references again. We have made various changes to make introduction more consistent, adding also one reference of IJERPH related to PWB. We have eliminated some references in Results, which we believe does not affect the integrity of the article itself. To sum up, the total number of references remains almost the same (49 now versus 50 before).

(II) Any revisions to the manuscript should be marked up using the “Track Changes” function if you are using MS Word/LaTeX, such that any changes can be easily viewed by the editors and reviewers. 

We have checked up all our corrections with the “Track Changes” tool in MS Word, as you recommended. Please notice: due to the corrections made, layout if the text and graphics have been modified.

(III) Please provide a cover letter to explain, point by point, the details of the revisions to the manuscript and your responses to the referees’ comments. 

We have elaborated this new cover letter to explain, point by point, the details of the revisions done and the responses to the referees’ comments, as you recommended.

(IV) If you found it impossible to address certain comments in the review reports, please include an explanation in your rebuttal.

The authors agree, but there has been no need to do so.

(V) The revised version will be sent to the editors and reviewers.

The authors agree with this rule.

Reviewer 2 Report

The authors aimed to assess if infection (according to infection after fracture fixation criteria, IAFF), immediate partial weight bearing (PWB) and/or the new GammaTScore can predict early cut-out in extracapsular hip fractures treated through intra medullary osteosynthesis.

Introduction must be shortened and include only very relevant information.

Please provide images representing GammaTScore parameters

Please define more in details early/not early cut out.

Infections should be detailed further (early, delayed, depth...)

How were cut out treated?

Limitations must be acknowledged

Author Response

Comments to the Review Report 2:

English language and style: Extensive editing of English language and style required 

In this regard, we have shortened and revised the whole text again with the help of a professional translator (for the second time) and improved its style, making it more attractive and easier to read, with the addition of 2 more figures and the removal of one table.

Questions:

Does the introduction provide sufficient background and include all relevant references? Must be improved.

The introduction has been shortened notably and the bibliographic citations have been revised (see below).

Are all the cited references relevant to the research? Yes

While we considered all citations relevant to the article, in deference to the other reviewer we have rechecked all references, altering some of them in order to make the document more consistent, which we believe does not affect the integrity of the article itself.

Is the research design appropriate? Can be improved

We agree with the reviewer that the research design could have been better, especially if we had had more early cut-out cases, but recruitment is tremendously difficult and hard in such an aging population. This circumstance has forced us to use statistical tests for non-normal samples such as the Mann-Withney U test in the study of our prognostic tool. However, the incidence of early cut-out in our study is within that reported in the literature, as we state in Discussion.

For more clarity, we include here the results of the normality tests for the variables shown in the subsequent bar charts, as a justification for the use of Mann-Whitney U test(p-value of K-S tests for normality).

Variable p (No Early Cut-out) / p (Early Cut-out)

-TAD 0.003 / 0.200

-Reduction 0.000 / 0.015

-Osteosynthesis 0.000 / 0.200

-Instability 0.000 / 0.182

As p<0.05 for the No Early cut-out group, the null hypothesis is rejected: there is no evidence that the data follow a normal distribution.

As p>0.05 for the Early cut-out group, the null hypothesis is not rejected: they follow a normal distribution (although the significance values are not very high).

However, the incidence of early cut-out in our study is within that reported in the literature, as we state in Discussion (less than 4%).

Our sample has an important follow-up and homogeneity, controlling confusion factors like osteoporosis, and our results are consistent with the new literature findings that support the role of IAFF and an increasingly important fracture reduction versus TAD, possibly because the latter is so implanted in surgeons that it is difficult to find relevant differences between cases and controls. 

Are the methods adequately described? Must be improved

Regarding this issue, the authors have preferred to shorten the text and refer the reader to reference number 33, which explain in detail how the clinical and radiological variables and follow-up were obtained. Nevertheless, key points have been pointed out like cut-out or introduced, like confirmatory major criteria for IAFF diagnosis.

Are the results clearly presented? Must be improved

The authors have made a considerable effort to organize the results and present them in an attractive way, minimizing the use of long tables and long paragraphs. The inclusion of discrete variables for GammaTScore marking have limited the use of other graphics, but our commitment to this system of punctuation facilitates an easy use of the tool for an immediate result, which it is really interesting in our busy clinical offices.

Are the conclusions supported by the results? Can be improved 

The Conclusions have been reformulated and shortened for better understanding and clarity for the reader.

Comments and Suggestions for Authors

Introduction must be shortened and include only very relevant information:

The introduction has been shortened by approximately 5 lines of text and the bibliographic citations have been revised. We have eliminated some references in Results, which we believe does not affect the integrity of the article itself. To sum up, the total number of references remains almost the same (49 now versus 50 before), except new recommendation by your revision.

Please provide images representing GammaTScore parameters:

Table 1 has been replaced by 2 explanatory figures on how to evaluate and obtain the partial (each parameter) and GammaTScore final mark (figures 2 and 3). 

Please define more in details early/not early cut out:

In Materials and Methods, we have corrected: "Early cut-out was defined as any grade of disrotation and/or migration of the PLD in the cephalic fragment, without (incomplete, grades I and II) or with (complete, grade III) articular damage, at any point throughout follow-up until 6 months postoperatively."

Infections should be detailed further (early, delayed, depth...):

All infections were considered depth infections and met at least one of the confirmatory criteria for IAFF:
1) Fistula, sinus or wound rupture (with communication to bone or implant).
2) Purulent drainage from the wound or presence of pus during surgery.
3) Phenotypically indistinguishable pathogens identified by culture from at least two separate deep tissue/implant (including sonication-fluid) specimens taken during an operative intervention.
4) Presence of microorganisms in deep tissue taken during an operative intervention, as confirmed by histopathological examination using specific staining techniques for bacteria or fungi.
Regarding the time of diagnosis of IAFF, Table 4 specifies in the "IAFF (weeks)" column the week in which it was detected (in the Discussion we also we also stated "Note that in 2 complicated cases IAFF occurred weeks later, which makes suspicion even more relevant".)

How were cut out treated?

In Table 4 we specify in the column "Consolidation/Reintervention (weeks)" the surgical treatment and the week in which reoperation was done (if needed), as well as the type of operation performed.

Limitations must be acknowledged

In Discussion, just before Conclusions, we said: "Our study also has some weaknesses, especially the small number of early cut-out cases, and its retrospective, non-randomized design that maybe limit  results and reliability"

Round 2

Reviewer 1 Report

The revised manuscript is greatly improved.

Reviewer 2 Report

In my opinion, the Authors made great efforts in the attempt to ameliorate their paper. it now merits publication.